# Monitoring Treadmill Physical Exercise and Recovery in Elite Water Polo Players with Local Muscle Oxygen Saturation Measurements—Regional and Sex Differences

**DOI:** 10.3390/jfmk10040464

**Published:** 2025-11-28

**Authors:** Máté Babity, Márk Zámodics, Éva Kovács, Ágnes Bucskó-Varga, Panka Kulcsár, Dóra Boroncsok, Regina Benkő, Alexandra Fábián, Márton Horváth, Dorottya Balla, Bálint K. Lakatos, Attila Kovács, Hajnalka Vágó, Béla Merkely, Orsolya Kiss

**Affiliations:** 1Heart and Vascular Center, Semmelweis University, 1122 Budapest, Hungary; babity.mate@semmelweis.hu (M.B.); zamodics.mark@semmelweis.hu (M.Z.);; 2Department of Sports Medicine, Semmelweis University, 1122 Budapest, Hungary

**Keywords:** NIR spectroscopy, elite athletes, exercise test, oxygen saturation, sex difference, water polo

## Abstract

**Background**: Despite numerous data on whole-body responses, we have less information about local muscular changes during physical exercise in athletes. Oxygen saturation (SmO_2_) changes in the working muscles follow phases of load and are useful, as local metabolism could influence physical fitness. **Methods**: A total of 100 asymptomatic elite water polo players (63% male, age: 17.2 (interquartile range: 16.1–18.9) years) were examined using near-infrared spectroscopy to measure SmO_2_ in both vastus lateralis and left deltoid muscles during continuous uphill running treadmill exercise. **Results**: Differences were observed between upper and averaged lower limb resting SmO_2_ (82.1% (77.0–89.0%) vs. 68.3% (59.2–73.6%), *p* < 0.001). During exercise, the relative decrease in averaged lower limb SmO_2_ was greater compared to the upper limb at the anaerobic threshold (−0.371 (−0.539–−0.200) vs. –0.224 (−0.340–−0.099), *p* < 0.001) and at maximal exercise (−0.557 (−0.750–−0.411) vs. –0.420 (−0.556–−0.271), *p* < 0.001). Higher averaged lower limb relative SmO_2_ was recorded compared to the upper limb after 5 min cool-down (+0.081% (−0.046–+0.195%) vs. –0.047% (−0.140–+0.000), *p* < 0.001). No differences were found between males and females in resting lower limb SmO_2_. Both sexes showed a monotonic decrease in SmO_2_ during exercise, with differences in the relative values at the anaerobic threshold and at maximal intensity. Females exhibited a rebound in SmO_2_ after a 5 min cool-down. **Conclusions**: We provide insights into SmO_2_ alterations during maximal-intensity exercise and recovery through the measurements of elite water polo athletes, also highlighting sex differences in SmO_2_. Measuring local SmO_2_ changes is a promising additional method in physical fitness follow-up.

## 1. Introduction

Several structural and functional cardiovascular parameters of sport adaptation have become an area of detailed research in recent decades. The results of these studies help to differentiate between physiological and pathological changes [1,2,3,4,5]. Besides focusing on screening for pathological signs, several data have become available showing a strong connection between cardiovascular sport adaptation signs and improvement of performance in sports [6,7,8,9].

Beyond short- and long-term cardiovascular adaptations to physical activity, several other factors and sport adaptation mechanisms that affect the entire human body also play a role in sport performance. To deliver oxygen from the air to the working muscles, athletes require appropriate lung function and circulatory redistribution, blood composition, as well as optimal condition of the locomotor muscles involving metabolic changes up to the level of enzymes and ionic channels [10,11,12]. To obtain optimal energy production for the muscles, local changes are essential to provide a sufficient oxygen supply. As part of these adaptation mechanisms, the blood hemoglobin (Hb)-oxygen affinity changes dynamically during exercise, decreasing in the working muscles due to changes in pH, carbon dioxide concentration, and temperature. Inside the muscle cells, myoglobin (Mb) holds and carries bound oxygen molecules [11]. In the case of optimal oxygen supply, the aerobic energy required for the working muscles is primarily provided by the mitochondria. Therefore, the quality and quantity of the mitochondria of the muscles also influence maximal oxygen uptake [13]. As a consequence of these complex mechanisms, local muscular oxygen saturation analysis could be a part of physical fitness measurements and follow-up.

Oxygen transport molecules have different absorption spectra in oxygenated and deoxygenated forms. However, absorption spectra of Mb and Hb are similar, thereby in vivo non-invasive differentiation between them is ambiguous. Unlike the light at the visible spectrum, the light in the infrared spectrum (~700–900 nm) is able to penetrate to human tissues. Based on the scattering nature of the tissues, near-infrared spectroscopy (NIRS) technology measures the oxygenation status of Hb in small vessels (<1 mm diameter), as well as Mb and the cytochrome oxidase enzyme in the skeletal muscle tissue. From the distance of the emitter and the detectors, the depth of the measurements and thus the studied tissues can be defined. The adipose tissue thickness could influence the NIRS measurements, but with proper mathematical models, these disturbances in signal detection could be avoided [14]. Since the contribution of cytochrome oxidase to the NIRS results is low, these measurements principally detect combined Hb and Mb oxygen saturation [15,16]. Although the determination of the absolute values of oxy[Hb + Mb] and deoxy[Hb + Mb] is not possible with continuous-wave NIRS technology, the ratio of these parameters can be obtained as mixed muscle tissue oxygen saturation (SmO_2_).

During physical activity, different muscles of the extremities and the trunk do not activate uniformly. For example, one of the most significant changes could be observed in the vastus group of the femoral muscle during uphill running [17]. Lateral vastus is a superficial muscle, covered by relatively thin adipose tissue, thereby suitable for reliable NIRS measurements.

In our research, the primary aim was to observe local SmO_2_ changes that develop in the contracting muscles before, during, and after increasing intensity uphill running in elite water polo players. We evaluated the SmO_2_ values as part of maximal-intensity cardiopulmonary exercise testing (CPET) and also compared these changes between male and female elite athletes.

## 2. Material and Methods

### 2.1. Participants

In this cross-sectional study, 100 asymptomatic Hungarian elite water polo players, both male and female, were examined from January 2021 to December 2022. All athletes were members of the youth or adult national water polo teams (Tier 4 or 5 according to McKay et al. [18]). All examined athletes were Caucasian. Our measurements were carried out as part of a detailed sports cardiology screening of asymptomatic athletes supervised by a specialist in sports cardiology. These examinations consisted of a detailed questionnaire regarding sports activity and medical history, personal consultation, resting 12-lead ECG, resting blood pressure measurement, blood sample examinations, bioimpedance-based body composition analysis, resting cardiac ultrasound, and maximal-intensity cardiopulmonary exercise testing. All members of the national water polo teams who underwent extended sports cardiology screening at our clinic were invited to participate in the study, and all of the examined athletes accepted the participation. Any signs of internal, cardiovascular, or musculoskeletal diseases, as well as major injuries in the last 6 months, were considered as exclusion criteria. Athletes who were suspended from regular physical activity in the last 6 months were also not invited to take part in the examinations. Due to these criteria, no exclusion was necessary for any athletes. Prior to the study, all participants and their legal guardian (in case of age <18 years) gave oral and written informed consent to the examinations and the research purposes after providing verbal information and answers to all arising questions. The Medical Research Council of Hungary approved the study (IV/10282-1/2020/EKU) in accordance with the Ethical Guidelines of the Helsinki Declaration and Good Clinical Practice.

### 2.2. Procedures

All measurements were performed at least 12 h after the last training sessions or matches [19]. To ensure similar conditions for all athletes, measurements took place in the daytime, in an air-conditioned laboratory with constant temperature and humidity in line with the instructions of the manufacturer.

#### 2.2.1. Cardiopulmonary Exercise Testing

Cardiopulmonary exercise testing was implemented on a treadmill ergometer (T-2100, GE Healthcare, Helsinki, Finland; Respiratory Ergostik, Geratherm, Geratal, Germany; Blue Cherry V1.3.3.1, Geratherm, Geratal, Germany) with an incremental protocol starting with a 2 min flat walk of 6 km/h, followed by continuous 8 km/h uphill running with an increasing slope of 1.5% every minute until exhaustion. Maximal intensity was considered to be achieved if the athlete reported maximal subjective exhaustion and either the respiratory exchange ratio (RER) was over 1.1, or flattening could be observed in the oxygen uptake and heart rate curves. After stopping running, measurements were continued during a 1 min walk at 4 km/h and a further 4 min resting period [20]. Blood lactate levels were measured from fingertip capillary blood drops at rest, during continuous exercise in every second minute, at maximal load, and at the fifth minute of the cool-down (Laktate Scout 4+, EKF Diagnostik, Barleben, Germany). Anaerobic threshold was determined based on the lactate levels (a sudden increase in lactate level, with the value around 4 mmol/L), increase in the respiratory gas exchange ratio over 1.0, as well as the kinetics of the recorded Wasserman graphs (e.g., a sudden increase in ventilation and carbon-dioxide production, crossing of the oxygen uptake and carbon-dioxide production curves) [21].

#### 2.2.2. Muscle Oxygen Saturation Measurements

Muscle oxygen saturation measurements were performed using a commercially available, validated NIRS device (Moxy, Fortiori Design LLC, Hutchinson, MN, USA) during cardiopulmonary exercise testing [14,22]. Several studies have investigated muscle oxygenation in the vastus lateralis and deltoid muscles; these examinations have consistently applied the SENIAM guidelines for sensor placement, facilitating comparability between different research projects. Applying the SENIAM guidelines ensures consistency and reproducibility with previous studies that have examined muscle oxygenation using similar protocols [23]. The devices were placed over the vastus lateralis muscles at two-thirds of the distance from the anterior spina iliaca to the lateral side of the patella, 3–4 cm laterally from the anterior midline, and over the medial deltoid muscle, between the acromion and the lateral epicondyle of the elbow at the greatest bulge of the muscle [14,24,25,26]. The devices were equipped with light shields and attached to the dried and cleaned skin with elastic adhesive tapes without compression to the skin or the muscles. The appropriate placement of the devices was supervised by a specialist in sports cardiology. The vastus lateralis was examined as a muscle intensively working during running, while the deltoid muscle plays a significant role as the principal abductor of the arm and the stabilizer of the glenohumeral joint; therefore, it undergoes a significant sports adaptation in water polo. However, the deltoid muscle was used as a control muscle that does not work intensively during running in our examinations. Averaged 10 s values of the SmO_2_ were analyzed at rest, at the anaerobic threshold, at maximal intensity exercise, and after 5 min of cool-down. Resting and cool-down measurements were performed in a chair sitting, resting position, while anaerobic threshold and maximal intensity measurements were performed during running. The lower limb measurements were performed simultaneously but separately on both vastus lateralis muscles. In water polo, both of the lower limbs are used similarly, thereby the proper determination of the dominant and non-dominant sides could be ambiguous. Therefore, for the comparisons, the average value of the right and left measurements was taken into account to eliminate the possible differences due to the laterality of the athletes. The relative changes were calculated as the difference from the resting measurements and the ratio of the difference to the resting value. All examinations and data collection were supervised by a cardiologist and sports medicine specialist.

### 2.3. Statistical Analysis

Descriptive statistical values are shown as number (percentage), or as median (interquartile range: 1st quartile–3rd quartile [IQR: Q1–Q3]) for continuous variables. Statistical analyses were performed using a dedicated software (IBM SPSS Statistics ver 28.0.1.0, IBM Corp, New York, NY, USA). The Shapiro–Wilk Test, Kurtosis and Skewness values, and visual assessments (histogram and Q–Q plot) were performed for testing the normality of the parameters. Due to the non-normality of some measured parameters, uniformly Wilcoxon Rank Sum Test (Mann–Whitney U Test), Wilcoxon Signed Rank Test or Related-Samples Friedman’s Two-Way Analysis of Variance by Ranks with Bonferroni correction were carried out for comprehensive statistical analysis. The threshold for statistical significance was defined as *p* < 0.05, while *p* < 0.001 indicates high statistical significance. All missing data were proved to be missing completely at random, thereby, available-case analysis was carried out for statistical evaluation.

## 3. Results

### 3.1. Study Population

We examined 100 elite water polo players (age: 17.2 years (IQR: 16.1–18.9 years), male: 63%). They trained a median of 17.5 h/week (IQR: 15.0–21.0 h/week). All examined athletes were players of the adult or youth national water polo teams of Hungary. The basic parameters of the examined athletes, divided by sex, are presented in Table 1.

### 3.2. Muscle Oxygen Saturation Values

First, SmO_2_ values were analyzed separately for each limb at each measurement point. Averaged values for the left and right vastus lateralis muscles were calculated, representing SmO_2_ of the lower limbs and eliminating the differences between the dominant and non-dominant legs. The summary of these measurements is depicted in Table 2. Raw data is available in Appendix A.

### 3.3. Muscle Oxygen Saturation Changes During Exercise

During the increasing slope uphill running, a monotonic decrease was measured in the SmO_2_ values. Regarding the averaged lower limb values, significant differences were observed between all measurement time points, except between the resting and cool-down values (Figure 1). In terms of the upper extremity measurements, a significant decrease could also be observed during the exercise. Although higher values were recorded after the cool-down compared to the measurements during exercise, saturation values remained significantly lower compared to the resting measurements (Figure 1).

### 3.4. Differences Between the Muscle Oxygen Saturation of the Upper and Lower Extremities

Significant differences were observed in the resting SmO_2_ values between the averaged lower and the upper extremities (respectively, 68.25% (IQR: 59.2–73.6%) vs. 82.1% (IQR: 77.0–89.0%), *p* < 0.001). As compared to the resting measurements, the relative decrease in the SmO_2_ values during exercise was significantly greater in the averaged lower limb results matched to the upper extremity values at the anaerobic threshold (respectively, −0.371 (IQR: −0.539–−0.200) vs. −0.224 (IQR: −0.340–−0.099), *p* < 0.001) and at maximal intensity exercise (respectively, −0.371 (IQR: −0.750–−0.411) vs. −0.420 (IQR: −0.556–−0.271), *p* < 0.001) as well. On the contrary, significantly higher relative differences in SmO_2_ values were recorded in the averaged lower extremity compared to the upper extremity in the cool-down (respectively, +0.081 (IQR: −0.046–+0.195) vs. –0.047 (IQR: −0.140–+0.000), *p* < 0.001). All results detailed above indicate high statistical significance.

### 3.5. Sex Differences in Muscle Oxygen Saturation Measurements

For both sexes separately, a monotonic continuous decrease could be observed during the exercise in SmO_2_ values, and significant differences were measured between resting, anaerobic threshold, and maximal intensity SmO_2_ values (Table 3, Figure 2). The resting and cool-down results were similar in male athletes, while female athletes showed a rebound effect (*p* < 0.05).

In case of the lower extremities, no significant sex differences were found in terms of the resting SmO_2_ values. Male water polo players showed greater muscle desaturation than females at the anaerobic threshold and at maximal intensity training (Figure 2).

Regarding the upper extremity, the resting SmO_2_ values proved to be higher in female athletes, but no sex differences were found in the relative ratio of exercise desaturations (Table 3). However, the cool-down relative changes in saturation values were significantly higher in female athletes.

## 4. Discussion

To the best of our knowledge, we have no previous data about SmO_2_ measurements of elite water polo athletes. The focus of the current study was the cross-sectional examination of the SmO_2_ changes in a large population of elite water polo players on a continuous uphill running protocol. The main findings in the present study regarding the uphill running SmO_2_ values of elite water polo players were as follows: (1) averaged lower limb and upper limb SmO_2_ values decrease significantly during exercise, (2) a greater desaturation could be observed in the working muscles of the lower extremities compared to the less involved upper extremities, (3) significant differences were found between SmO_2_ values of male and female athletes, (4) quick return of muscle oxygen saturation parameters to the resting values or, in case of female athletes, also a rebound effect could be observed in the 5 min cool-down period.

The physiological changes behind the aforementioned results could be explained as follows. During exercise, the oxygen demand of the contracting muscles increases. As a result of the circulatory redistribution, the blood flow of the contracting skeletal muscles increases significantly. The oxygen extraction capacity and the blood transit time also increase in well-trained muscles during exercise. As a result, the blood supplying the muscles with oxygen becomes more desaturated compared to the resting state; therefore, the mixed muscle oxygen saturation values decrease [27,28,29]. During the cool-down phase after exercise, muscle blood flow may remain elevated for a period to eliminate metabolic substances and repay the oxygen debt [30,31]. In our study, we also observed this phenomenon after 5 min of recovery in the case of the upper limbs, but not in the lower limbs of female athletes, presumably due to the markedly increased circulation and fast recovery of the lower limbs.

Previous studies and meta-analyses confirmed that the anaerobic threshold could be determined by SmO_2_ evaluation with moderate to good reliability; on the contrary, in some cases, the SmO_2_ values did not show correlations with other CPET results [32,33,34]. Prior examinations with small case numbers have demonstrated a decrease in muscle oxygen saturation due to exercise as well as a negative correlation between muscle oxygen saturation values and oxygen consumption [24,35,36,37,38]. Based on a previous study with machine learning models, the VO2 values could be predicted reliably from SmO_2_ values obtained from the right and left vastus lateralis muscles and heart rate [39]. The differences between previous results can be explained by different types of studied sport, different sport intensity of the participants, different exercise examination protocols, and different examined muscles. Although we already have some scientific data about the changes in SmO_2_ during physical activity, no results specific to water polo players are available in the literature. A more profound analysis of different sport types could lead to a deeper understanding of the unique local changes in the working muscles during physical activity.

In a previous study, runners were examined during hilly terrain running. The SmO_2_ values showed a better correlation with the changes in the terrain than the oxygen uptake or the heart rate, as these latter parameters showed a significant delay in response to the changes in the terrain [37].

In another study, the authors examined the SmO_2_ values among a young adult population. A different NIRS device was used, and the left lateral gastrocnemius muscle was examined on a flat, increasing speed running treadmill protocol (n = 23). The SmO_2_ values were 43.8 ± 14.7% at the lactate threshold, while 39.7 ± 14.0% at the peak of exercise. In the same study, the authors also examined the vastus lateralis muscle of cyclists on a cycling ergometer (n = 21), where the SmO_2_ values were 39.6 ± 17.1% at the lactate threshold and 26.2 ± 16.0% at maximal effort [40]. These results, examining the same muscles as our study, are comparable to our observations, although performing different sports activities. These similarities could suggest that the SmO_2_ measurements are a useful and comparable method for a wide range of athletes, regardless of the environment of the exercises.

A swimming field measurement carried out among swimmers found greater desaturations in the deltoid muscle compared to our results, as the deltoid muscle and the upper body are more strenuously involved during swimming compared to treadmill running [41]. These differences underline that, in the case of SmO_2_ measurements, it is essential not only to compare the same muscles of the athletes but also to consider the activation level of certain muscles, as physical activity greatly influences the measurements.

It is also important to note that the level of the athlete, and therefore the perceived physical load, could influence the SmO_2_ values. In a previous study, the authors compared the SmO_2_ values of national-level swimmers to regional-level swimmers during a field test. In this research the national-level athletes achieved lower SmO_2_ values for a longer period of time at the same test [42].

### 4.1. Comparison Between the Upper and Lower Extremity SmO_2_ Measurements

In our study, we verified that the desaturation during running is more pronounced in the lower limbs than in the upper extremities, since the work of the legs is more pronounced, and their oxygen demand is greater. At the same time, the mixed muscle oxygen saturation measured above the deltoid muscle also decreased significantly, as the upper limbs also work during running, although with lower intensity than the legs.

In previous case reports, the authors suggest that some muscles were unaffected by the central fatigue mechanisms during cross-country skiing, while other regions were more affected by fatigue during the race. These results suggest that the examination of a particular part of the skeletal muscles could provide more detailed information about the activation of specific muscle groups during exercise, as well as it could highlight the individual characteristics of an athlete regarding sports activity and recovery [43,44]. These results align with our findings, as the relative desaturation during the running test of the less involved upper extremity was lower than the lower extremity in our study. Further analysis should be performed during regular exercises in water polo, as well as to confirm the previous concept of fatigue mechanisms in water team sports.

In another study examining swimmers and triathletes during a 200 m freestyle swimming test, no significant difference was found in the tissue saturation index between the vastus lateralis and latissimus dorsi muscles of the swimmers; however, a difference was observed between the above muscles among triathletes. Presumably, swimmers used their upper and lower body muscles to a similar extent, while the triathletes predominantly used their upper body during the same swimming test [45]. These results suggest that differences in how individual athletes complete the same exercise could be highlighted with these measurements. Therefore, their strengths and weaknesses could be determined, which would aid in further training planning and improvement.

### 4.2. SmO_2_ Value Changes Regarding the Sexes

We assumed that the resting SmO_2_ values of the upper and lower extremities should be similar between male and female athletes, which proved to be true, considering the averaged lower limb results. In the case of the upper limb, the female athletes had higher resting SmO_2_ values. During the continuous uphill running, the male water polo players had lower SmO_2_ values in both upper and lower extremities compared to the female athletes. Moreover, in female water polo players, a rebound effect was observed after the 5 min cool-down, compared to the resting SmO_2_ values in the lower limb; in contrast, similar SmO_2_ values were observed in male athletes at these time points.

According to a previous publication carried out with the same SmO_2_ measurement device in a slightly older population, a greater decrease in the oxygen saturation of the vastus lateralis muscle was observed in the case of male athletes on a bicycle ergometer as a result of exercise (male: rest: 65 ± 10%, anaerobic threshold: 34 ± 18%, maximal intensity: 31 ± 19%; female: rest: 57 ± 14%, anaerobic threshold: 44 ± 12%, maximal intensity: 41 ± 11%). These results are similar to our study, as we observed greater desaturation of the vastus lateralis muscle at the anaerobic threshold and at the peak of exercise in males compared to females. Interestingly, no significant further desaturation was measured between the anaerobic threshold and the maximal intensity in this study in contrast to our measurements. Moreover, a greater degree of desaturation was measured in the intercostal muscles in female athletes [46]. In another study, also conducted on a bicycle ergometer among young adult athletes, the authors found lower SmO_2_ values in male cyclists compared to females, both in the working and stabilizing muscles [47]. These sex differences in SmO_2_ values are consistent with many previous data referring to more pronounced sport adaptation changes in male athletes. The results may be linked to sex differences in local metabolic processes, different distribution of working muscles during the same activity, as well as the regularly higher maximal exercise capacity of male athletes. On the other hand, presumably due to hormone-mediated vasodilatation, improved vasoactive function could also be observed in female athletes, suggesting less intensive desaturation is needed to meet the needs of the muscles at the same perceived loads [48,49].

### 4.3. Limitations

As a limitation, it is essential to note that this measurement method is not suitable for directly measuring the blood flow of muscles. Thus, it is not suitable for the numerical determination of the amount of delivered oxygen. This technique can only determine the relative ratio of oxygenated and deoxygenated carrier molecules. As this methodology is relatively simple, it can be used in everyday practices beyond laboratory settings.

Moreover, we did not have data on the thickness of the subcutaneous fat tissue at the time of the examinations, which may have slightly influenced the measurements. As all the examined athletes were elite players, no significant amount of subcutaneous fat tissue disturbing the measurements is presumable.

Additionally, a limitation of our study was that the statistician could not be blinded due to the nature of the data analysis process.

It would also be worthwhile to conduct repeated examinations within the framework of the research, allowing for a longitudinal follow-up on the changes in the athletes.

## 5. Conclusions

According to the results of the current study, significant desaturations occur during uphill running in both of the vastus lateralis and in the deltoid muscles among elite water polo players, and these changes are more pronounced in the lower extremities. The resting SmO_2_ values were similar in male and female athletes, while greater desaturation was observed among male athletes during uphill running in the vastus lateralis muscle. A rebound effect in the SmO_2_ values could be registered after a 5 min cool-down in female athletes compared to the resting values, while similar results were measured at rest and after the cool-down in male athletes, referring to the quick recovery of local muscular oxygen metabolism following a short-term maximal intensity exercise test. Studying SmO_2_ as part of physical fitness monitoring, training, and optimal recovery planning could provide valuable insights into local metabolism.

## Figures and Tables

**Figure 1 jfmk-10-00464-f001:**
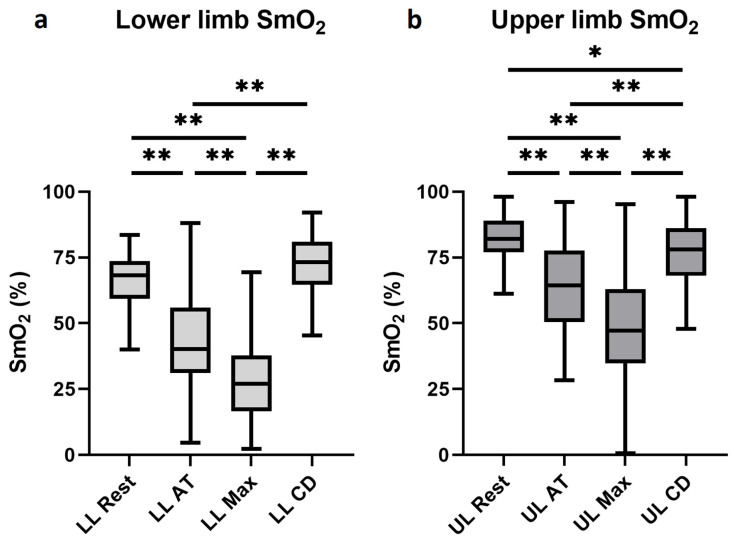
Averaged left and right lower extremity and upper extremity SmO_2_ values. Averaged left and right lower extremity SmO_2_ values are illustrated on panel (**a**), while the upper extremity SmO_2_ values are on panel (**b**), at rest, at anaerobic threshold, at maximal intensity, and at cool-down. Abbreviation: SmO_2_, muscle oxygen saturation; LL, lower limb; UL, upper limb; AT, anaerobic threshold; Max, maximal intensity; CD, cool-down; *: *p* < 0.05; **: *p* < 0.001.

**Figure 2 jfmk-10-00464-f002:**
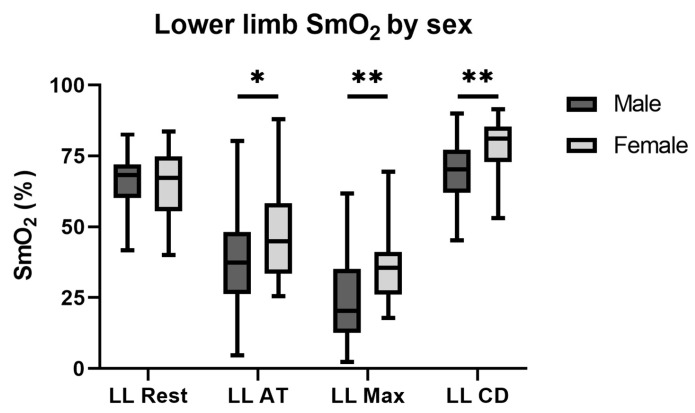
Lower extremity SmO_2_ at rest, anaerobic threshold, maximal intensity, and after a 5 min cool-down by sex. The resting muscle oxygen saturations do not differ between the two sexes in the case of the averaged lower extremity measurements. Male athletes showed a greater desaturation during exercise compared to female athletes. After 5 min of cool-down, the female athletes showed higher SmO_2_ values than the males. Abbreviation: SmO_2_, muscle oxygen saturation; LL, lower limb; AT, anaerobic threshold; Max, maximal intensity; CD, cool-down; *: *p* < 0.05; **: *p* < 0.001.

**Table 1 jfmk-10-00464-t001:** The basic characteristics of the examined elite water polo players.

	All Players	Male	Female	*p*
Participant [n] (%)]	100 (100%)	63 (63%)	37 (37%)	-
Age [year]	17.2 (16.1–18.9)	17.6 (16.2–19.0)	17.0 (15.1–18.6)	0.237
Training [h/week]	17.5 (15.0–21.0)	17.5 (15.0–21.0)	17.5 (14.5–22.9)	0.816
Height [cm]	183.0 (173.8–189.3)	187.0 (183.0–193.0)	173.0 (171.0–178.0)	**0.001**
Weight [kg]	77.0 (70.0–88.0)	83.0 (75.5–92.0)	72.0 (67.0–77.0)	**0.001**
BMI [kg/m^2^]	23.5 (22.1–25.3)	23.4 (22.2–25.7)	23.7 (22.1–25.1)	0.631

Abbreviation: BMI, body mass index. The *p* values refer to the comparisons between the sexes, with bold the significant changes are highlighted.

**Table 2 jfmk-10-00464-t002:** The muscle oxygen saturation values of the left and right vastus lateralis muscles and of the left deltoid muscle.

	Left Vastus Lateralis	Right Vastus Lateralis	Average of Left and Right Vastus Lateralis	Left Deltoid Muscle	*p*
Rest [%]	65.0 (57.1–72.0)	70.1 (59.2–77.6)	68.3 (59.2–73.6)	82.1 (77.0–89.0)	**<0.001**
Anaerobic threshold [%]	39.0 (29.0–54.0)	40.0 (29.6–56.6)	40.2 (31.1–56.0)	64.3 (50.4–77.7)	**<0.001**
Maximal intensity [%]	27.0 (17.1–36.6)	25.9 (15.7–40.0)	27.0 (16.6–37.7)	47.2 (34.7–62.9)	**<0.001**
5 min cool-down [%]	71.8 (61.4–78.0)	77.1 (65.0–83.1)	73.3 (64.7–80.9)	78.0 (68.0–86.1)	**0.003**

The *p* values refer to the comparisons between the averaged lower extremity- and the upper extremity values, with bold the significant changes are highlighted. The threshold for statistical significance was defined as *p* < 0.05, while *p* < 0.001 indicates high statistical significance.

**Table 3 jfmk-10-00464-t003:** Comparison of upper and averaged lower extremities muscle oxygen saturation values and relative changes to resting measurements by sexes.

	Male	Female	*p*	Relative Changes Male	Relative Changes Female	*p*
Averaged lower extremities						
Rest	68.3% (60.2–72.0%)	67.2% (55.5–74.9%)	0.960	0	0	-
Anaerobic threshold	37.4% (26.3–48.2%)	44.9% (33.6–58.4%)	**0.024**	−0.441 (−0.635–−0.229)	−0.291 (−0.411–−0.177)	**0.003**
Maximal intensity	20.3% (12.6–35.2%)	35.5% (26.1–41.1%)	**<0.001**	−0.669 (−0.808–−0.469)	−0.457 (−0.549–−0.358)	**<0.001**
5 min cool-down	70.3% (62.0–77.2%)	78.3% (72.0–85.0%)	**<0.001**	+0.022 (−0.090–+0.127)	+0.164 (+1.023–+0.291)	**<0.001**
Upper extremity						
Rest	80.0% (74.0–84.0%)	87.8% (83.0–92.0%)	**<0.001**	0	0	-
Anaerobic threshold	56.4% (48.2–72.0%)	73.8% (57.8–82.6%)	**0.007**	−0.245 (−0.345–−0.111)	−0.176 (−0.319–−0.055)	0.268
Maximal intensity	41.3% (31.2–56.1%)	53.6% (40.2–68.8%)	**0.009**	−0.444 (−0.572–−0.298)	−0.374 (−0.543–−0.200)	0.127
5 min cool-down	73.1% (63.6–80.2%)	86.0% (78.0–94.4%)	**<0.001**	−0.076 (−0.165–−0.011)	−0.011 (−0.077–+0.007)	**0.009**

The relative values refer to the changes compared to the resting measurements, expressed as a ratio to the resting values. The *p* values refer to the comparisons between sexes at each time points, with bold the significant changes are highlighted. The threshold for statistical significance was defined as *p* < 0.05, while *p* < 0.001 indicates high statistical significance.

## Data Availability

The original contributions presented in this study are included in the article/Appendix A. Further inquiries can be directed to the corresponding author.

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
