# Peer review of "Monitoring Treadmill Physical Exercise and Recovery in Elite Water Polo Players with Local Muscle Oxygen Saturation Measurements—Regional and Sex Differences"

_jfmk, 2025, doi:10.3390/jfmk10040464_

Round 1

Reviewer 1 Report

Comments and Suggestions for Authors

The authors present an interesting study investigating oxygen saturation changes of activated muscles during exertion effort and rest in elite national-level water polo athletes. This generally seems to be a well-executed and well-described study, but there are some areas that could use attention or clarification. Placement of NIR sensors on the deltoid was not described; the authors only stated that the procedure was developed in-house. The development and procedure should probably be written as a stand-alone paper or at least be detailed clearly in the current manuscript proposal. Was the statistician blinded? This should be stated, and if not, this is a potential limitation. There was no mention that dominant limbs were determined or how this was done; both need to be explicitly stated. Rather than being a prospective study, this seems to be more appropriately described as a cross-sectional study to this reviewer.  Overall, this reviewer  found the manuscript proposal novel and interesting and looks forward to reading a revision.

Comments on the Quality of English Language

Some proofreading and correction of style, grammar, and word use is needed, specifically on page 8, line 252, "upper mentioned results" should be "aforementioned results" or "above-mentioned results".

Author Response

The authors present an interesting study investigating oxygen saturation changes of activated muscles during exertion effort and rest in elite national-level water polo athletes. This generally seems to be a well-executed and well-described study, but there are some areas that could use attention or clarification.

Thank you for the comprehensive review. We really appreciate the time and effort taken in considering our article! We corrected all the assigned points and reworked the article according to your instructions. Your comments have significantly improved the quality of the manuscript; we hope it will meet your expectations for publication with the corrections. The corrections are highlighted in the article. Our answers are as follows:

1) Placement of NIR sensors on the deltoid was not described; the authors only stated that the procedure was developed in-house. The development and procedure should probably be written as a stand-alone paper or at least be detailed clearly in the current manuscript proposal.

Thank you for the suggestion. We extended the section with a more detailed description of the method section. We extended with the baseline guideline witch is followed in the literature and also guided our decision of the lower- and upper limb measurement points. P3 L132-140.

2) Was the statistician blinded? This should be stated, and if not, this is a potential limitation.

Unfortunately, in our settings, we did not have the opportunity to blind the statistician. We added this statement to the Limitations section. P10 L380-1

3) There was no mention that dominant limbs were determined or how this was done; both need to be explicitly stated.

Since in water polo both of the lower limbs are used similarly, thereby proper determination of the dominant and non-dominant sides could be ambiguous. The dominant side could be better described in the upper limb. For our comparisons, the averaged values of the lower-limb measurements were taken into account to filter out the minor differences in the lower limb measurements. We extended the manuscript at P4 L153-5.

4) Rather than being a prospective study, this seems to be more appropriately described as a cross-sectional study to this reviewer.

We used the terminology of “prospective”, because the data collection was prospective and not retrospective, however the measurements were cross-sectional in our population. For accuracy we changed “prospective” to “cross-sectional” in the manuscript. P2 L86, P8 L255

5) Overall, this reviewer found the manuscript proposal novel and interesting and looks forward to reading a revision. Some proofreading and correction of style, grammar, and word use is needed, specifically on page 8, line 252, "upper mentioned results" should be "aforementioned results" or "above-mentioned results".

Thank you for the comment, we revised the manuscript for stylistic and grammatical errors, and we corrected them.

Reviewer 2 Report

Comments and Suggestions for Authors

I had a opportunity to review the manuscript entitled “Monitoring treadmill physical exercise and recovery in elite 2 water polo players with local muscle oxygen saturation meas- 3 urements; regional and sex differences”.

Please find two minor comments

  1. “In this prospective study, 100 asymptomatic Hungarian elite water polo players from 86 both sexes were examined, from 2021 January until 2022 December”. Please describe in more detail how the final sample size was arrives (inclusion/exclusion criteria).
  2. Numbers, please use “.” Instead of “,”.

Author Response

I had an opportunity to review the manuscript entitled “Monitoring treadmill physical exercise and recovery in elite water polo players with local muscle oxygen saturation measurements; regional and sex differences”.

Thank you for the revision, we really appreciate the time and effort taken in considering our article! We corrected the assigned points and corrected the article according to your instructions. The corrections are highlighted in the article. Our answers are as follows:

Please find two minor comments

1) “In this prospective study, 100 asymptomatic Hungarian elite water polo players from both sexes were examined, from 2021 January until 2022 December”. Please describe in more detail how the final sample size was arrives (inclusion/exclusion criteria).

Thank you for the comment. We have extended the inclusion and exclusion criteria in the manuscript. Altogether, there was no need for exclusion in our study, which is now stated in the manuscript. P3 L95-101

2) Numbers, please use “.” Instead of “,”.

Thank you for your comment. We have corrected the decimal points in P3 L126 and P5 Table1.

Reviewer 3 Report

Comments and Suggestions for Authors

The authors' choice of topic and the execution of their research are remarkable in that they examined representatives of a sport that has not yet been covered in the relevant literature.
In addition, they examined a particularly large number of elite water polo players in a country where, to my knowledge, both men's and women's water polo are among the best in the world.

If the authors make corrections based on my comments and questions below, I support the manuscript’s publication in J. Funct. Morphol. Kinesiol.

line 18: 'Interquartile range' should be written out in full before the abbreviation IQR .

line 22: 59,0%? In Table 2 there is 59,2%.

It would be necessary to check all data showing relative changes appearing in the abstract, text, and tables. In my opinion they are not correct. However, it is possible that the values of relative changes were calculated differently than I did. If the authors have used any special calculations, it is worth describing them in the Methods section. Furthermore, it is confusing that the relative changes are not presented in a uniform manner. They are presented as percentages in the Abstract, but in the text of the Results and in Table 3, they are presented as decimals.

line 89: I suggest using the term "Caucasian" instead of "white."

line 97 and 102: It is unnecessary to repeat to mention the informed consent.

The data on the right side of Table 2 are demonstrated again in Fig. 1, and the data on the left side of Table 3 are represented again in Fig. 2 .  I think it is unnecessary to present the same results in both tables and figures. 

The upper limb data are missing from Fig.2 , although it is referenced in the text (line 234).

lines 220-221 and 242-243: The explanatory text is unnecessary for interpreting the boxplot diagram.

Fig. 2: The positions of the * and ** symbols do not correspond to the p values ​​in Table 3.

The attached supplementary file is unnecessary.

Author Response

The authors' choice of topic and the execution of their research are remarkable in that they examined representatives of a sport that has not yet been covered in the relevant literature.
In addition, they examined a particularly large number of elite water polo players in a country where, to my knowledge, both men's and women's water polo are among the best in the world.

If the authors make corrections based on my comments and questions below, I support the manuscript’s publication in J. Funct. Morphol. Kinesiol.

Thank you very much for your thorough review and positive feedback. We greatly appreciate your kind remarks regarding the representation and competitive excellence of Hungarian men’s and women’s water polo. We have made the requested changes in the manuscript and provide our detailed responses below.

line 18: 'Interquartile range' should be written out in full before the abbreviation IQR.

We have written out the ‘interquartile range’ in full the abstract in L18.

line 22: 59,0%? In Table 2 there is 59,2%.

Thank you for the comment. We have checked the results again and corrected the data in L22. We also changed the order of the lower and upper extremity values in L207-209 to represent the results more uniformly.

It would be necessary to check all data showing relative changes appearing in the abstract, text, and tables. In my opinion they are not correct. However, it is possible that the values of relative changes were calculated differently than I did. If the authors have used any special calculations, it is worth describing them in the Methods section. Furthermore, it is confusing that the relative changes are not presented in a uniform manner. They are presented as percentages in the Abstract, but in the text of the Results and in Table 3, they are presented as decimals.

Thank you for your valuable comment. To enhance clarity, we have standardized the presentation of relative changes to decimal format and corrected the values in Table 3 and throughout the manuscript. We have also added a brief explanation of our calculation method in the 'Methods' section L158-159. The relative changes were calculated as the difference from the resting measurement divided by the resting value.

line 89: I suggest using the term "Caucasian" instead of "white."

We have changed the wording in L90.

line 97 and 102: It is unnecessary to repeat to mention the informed consent.

Thank you for the comment. We have removed the redundant mention of informed consent in 98.

The data on the right side of Table 2 are demonstrated again in Fig. 1, and the data on the left side of Table 3 are represented again in Fig. 2.  I think it is unnecessary to present the same results in both tables and figures. 

Thank you for the suggestion. We chose to present the data both in tables and figures because visualizing the results facilitates interpretation. Additionally, the graphs display the minimum and maximum values, which complement the numerical tables. Presenting the exact data only in text could make the manuscript harder to follow; therefore, both formats were included to improve clarity and comprehension.

Also, in the case of Table 2, the SmO2 values at different locations were compared at the same time, while in Figure 1, the SmO2 values at the same locations were compared at different stages of the exercise.

The upper limb data are missing from Fig.2 , although it is referenced in the text (line 234).

We did not include the upper extremity data in the figure because a significant difference was found between the resting SmO2 values of male and female athletes. This difference persisted during exercise tests, but when comparing relative changes, it disappeared except for the cool-down phase. Including these absolute values could distract readers. The reference in L234 was added inadvertently; we have removed the corresponding text to avoid confusion.

lines 220-221 and 242-243: The explanatory text is unnecessary for interpreting the boxplot diagram.

Thank you for the recommendation. We have removed the boxplot interpretations from the captions.

Fig. 2: The positions of the * and ** symbols do not correspond to the p values ​​in Table 3.

Thank you for the remark. We have corrected the symbols in Figure 2 to align with the p-values reported in Table 3. Additionally, we reviewed our calculations and made the necessary corrections.

The attached supplementary file is unnecessary.

We have attached the raw data to promote transparency in our research. Additionally, we added it to facilitate further analysis of other workgroups, specifically measuring muscle oxygen saturations. Furthermore, to the best of our knowledge, the MDPI promotes data sharing concerning the minimal dataset that supports the findings of the studies.

Round 2

Reviewer 1 Report

Comments and Suggestions for Authors

This reviewer appreciates the authors' careful revision of their manuscript proposal and looks forward to reviewing a revised version.

Author Response

This reviewer appreciates the authors' careful revision of their manuscript proposal and looks forward to reviewing a revised version.

Thank you for the remarks. We have made the figures and tables clearer and also improved the grammar of the manuscript. For clarity, we have adjusted the coloring of Figure 1 to be more uniform and corrected the symbols in Figure 2. We have added borders to Table 1 to improve its readability.